# FROM BIAS TO BALANCE: EXPLORING AND MITIGATING SPATIAL BIAS IN LVLMS

## ABSTRACT

Large Vision-Language Models (LVLMs) have achieved remarkable success across a wide range of multimodal tasks, yet their robustness to spatial variations remains insufficiently understood. In this work, we present a systematic study of the spatial bias of LVLMs, focusing on how models respond when identical key visual information is placed at different locations within an image. Through a carefully designed probing dataset, we demonstrate that current LVLMs often produce inconsistent outputs under such spatial shifts, revealing a fundamental limitation in their spatial-semantic understanding. Further analysis shows that this phenomenon originates not from the vision encoder, which reliably perceives and interprets visual content across positions, but from the unbalanced design of position embeddings in the language model component. In particular, the widely adopted position embedding strategies, such as RoPE, introduce imbalance during cross-modal interaction, leading image tokens at different positions to exert unequal influence on semantic understanding. To mitigate this issue, we introduce **Balanced Position Assignment (BaPA)**, a simple yet effective mechanism that assigns identical position embeddings to all image tokens, promoting a more balanced integration of visual information. Extensive experiments show that BaPA enhances the spatial robustness of LVLMs without retraining and further boosts their performance across diverse multimodal benchmarks when combined with lightweight fine-tuning. Further analysis of information flow reveals that BaPA yields balanced attention, enabling more holistic visual understanding.

## 1 INTRODUCTION

Large Vision-Language Models (LVLMs) have achieved remarkable success across a wide range of multimodal tasks, including visual question answering (Alayrac et al., 2022; Xu et al., 2025), image captioning (Li et al., 2023; Lu et al., 2025), and open-ended reasoning (Zhu et al., 2025b; Wang et al., 2025). By combining powerful vision encoders with large language models (LLMs), these systems are able to integrate information from both modalities and perform complex reasoning. Despite these advances, LVLM still exhibits fundamental limitations when it comes to spatially robust semantic understanding of visual content (Imam et al., 2025; Li et al., 2025).

Recent efforts have begun to explore the *spatial bias* of LVLMs, motivated by its hypothesized connection to object hallucination. Xing et al. (2024) show that the widely used Rotary Position Embedding (RoPE) (Su et al., 2024) introduces a long-term decay effect (Peng et al., 2024), which impedes LVLMs from effectively capturing visual cues located linearly far from text tokens. In contrast, Zhu et al. (2025c) argue that such findings lack generalizability across architectures and introduce two novel attention calibration mechanisms to rectify spatially unbalanced attention with LVLMs. While insightful, their exploration of spatial bias remains confined to the context of object hallucination—a phenomenon that may also be influenced by other confounding factors, such as the inherent hallucination tendencies of LLMs. Consequently, a rigorous examination of spatial bias grounded in the fundamental aspect of *semantic understanding* remains largely unexplored. Moreover, their analyses focus primarily on *attention distributions*, failing to delve into the underlying causes of spatial bias in LVLMs.

To fill this gap, this work conducts a systematic investigation into the spatial robustness of LVLMs' semantic understanding when subjected to positional variations of critical visual information.

Specifically, we construct a probing dataset designed for image-text matching, where the same semantic content is placed in various spatial locations. The results show that LVLMs are highly sensitive to these alterations, often producing inconsistent or even contradictory outputs under such shifts. This phenomenon highlights a critical weakness in the spatial understanding capability of current LVLMs, indicating that their integration of image features is not position-invariant but biased toward certain preferences. Meanwhile, the observed spatial position preferences of LVLMs further demonstrate that the bias should not be attributed to the long-term decay property of RoPE.

To uncover the root cause of this vulnerability, we first investigate whether the vision encoder is responsible for the observed spatial bias. Our eraser search (Li et al., 2016; De Cao et al., 2020) experiments on its perceptual ability confirm that it consistently perceives visual features of key content, regardless of the spatial position of the key image. Building on this, we then analyze whether the encoder's semantic understanding is affected by spatial position. The high and stable similarity between text embeddings and visual features across different locations demonstrates that semantic encoding is also robust to spatial variation. These observations rule out the vision encoder as the source of spatial bias, suggesting that the issue originates in the LLM portion of the LVLM, where the visual features are processed for multimodal reasoning.

Based on the above observations, we hypothesize that spatial bias stems from the *imbalance in cross-modal interactions*, primarily introduced by position embeddings within the LLM backbone of LVLMs. Current models typically adopt RoPE (Su et al., 2024) or related schemes, which modulate attention scores between tokens based on their relative distances in the sequence. While this design has proven highly effective in unimodal text modeling, it becomes problematic in multimodal contexts. Specifically, all image tokens should *contribute equally* during cross-modal fusion with text tokens. The sequential distance bias inherent in RoPE disrupts this equitable interaction, leading to distorted cross-modal interaction and ultimately undermining the spatial robustness of LVLMs.

To address this, we propose a simple yet effective **Balanced Position Assignment (BaPA)** mechanism which assigns identical positional embeddings to all image tokens. This modification explicitly promotes a more balanced and thorough integration of visual cues in cross-modal interactions. Empirical results on our probing dataset show that applying BaPA *without retraining* leads to more balanced performance and remarkably higher accuracy. We further adapt BaPA for broader downstream tasks. Results on five benchmark datasets show that BaPA can enhance LVLMs across diverse multimodal benchmarks, demonstrating the effectiveness and generalizability of BaPA.

To summarize, our contributions are threefold. First, we provide a systematic investigation of spatial robustness in LVLMs's semantic understanding, empirically demonstrating through novel probes that LVLMs' predictions vary significantly with the spatial location of identical visual content. Second, we pinpoint the root cause of this bias to the LLM's position-sensitive cross-modal interactions, rather than to deficiencies in the vision encoder. Third, we introduce Balanced Position Assignment (BaPA), a lightweight and generalizable method that improves spatial robustness without sacrificing downstream performance, validated through both probing tasks and five multimodal benchmarks.

## 2 RELATED WORK

**Large Vision-Language Models.** LVLMs combine both visual and textual inputs, providing a more comprehensive understanding of visual spatial relationships, objects, and scenes (Bordes et al., 2024). Existing LVLMs typically comprise a visual encoder (Dosovitskiy et al., 2021; Radford et al., 2021), a projector (Alayrac et al., 2022), and a pre-trained LLM (Touvron et al., 2023). Through pre-training with image-text pairs and fine-tuning with preference or instruction, current LVLMs, like LLaVA (Liu et al., 2023a) and Qwen2.5-VL (Bai et al., 2025), have been successful in dialogue (Zhu et al., 2025a), question answering (Zhu et al., 2023), and complex reasoning (Zhu et al., 2025b). Nonetheless, LVLMs still exhibit numerous biases (Ruggeri et al., 2023; Wang et al., 2024a; Zhang et al., 2025b), which diminish the trustworthiness of their response. In this paper, we systematically investigate the under-explored issue of spatial bias of LVLMs and introduce a mitigation strategy from the perspective of positional balance.

**Position Encoding in Transformers.** Since Transformers (Vaswani et al., 2017) lack a natural understanding of sequence order, numerous studies have proposed various position encoding methods. Early works adopt sinusoidal absolute position encoding (Vaswani et al., 2017) and learnable em-

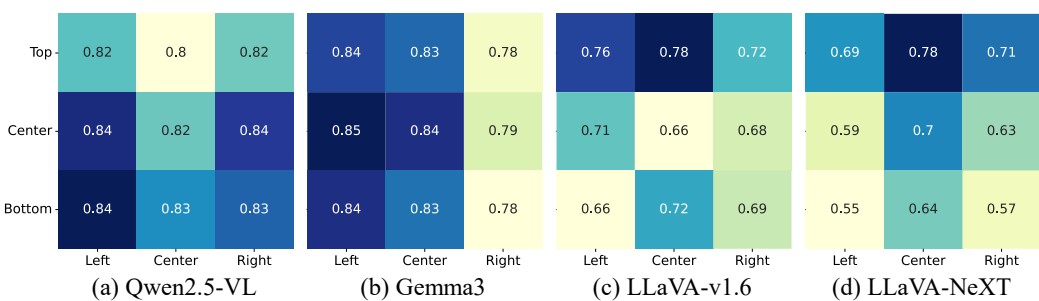

Figure 1: Results on our probe dataset, where each cell reports the model's accuracy when the key image $I_m$ is placed at a specific grid location $n$ in the $3 \times 3$ composite image $\boldsymbol{I}_{m,n}$.

beddings (Dosovitskiy et al., 2021), while later approaches focus on learnable and relative position encodings (Shaw et al., 2018; Huang et al., 2020; He et al., 2021; Ke et al., 2021), which offer more flexibility in capturing temporal dependencies. Among them, Rotary Position Embedding (RoPE) (Su et al., 2024) encodes relative positions through rotation matrices and has shown strong effectiveness in LLMs and LVLMs. Meanwhile, work on extensions to RoPE is emerging, such as LongRoPE (Ding et al., 2024), Unified RoPE (Wu et al., 2025), and CARoPE (Veisi et al., 2025). Despite this, current work find that the impact of RoPE leads to position bias in LLMs (Zhang et al., 2024; Chen et al., 2023), resulting in a phenomenon known as "lost-in-middle" (Liu et al., 2023b). Recently, Xing et al. (2024) investigate how RoPE affects object hallucination in LVLMs and propose a novel position alignment method to mitigate the long-term decay in RoPE. In contrast, we thoroughly explore the spatial bias grounded in the fundamental semantic understanding task and observe that the underlying issue lies not in long-term decay but in unbalanced position assignment of LVLMs. Although our approach also involves modifying the position encoding, it is motivated by a different rationale, and it is empirically more effective than Xing et al. (2024) on multiple benchmarks and even achieves balanced results without retraining.

## 3 PROBING TASK FOR SPATIAL ROBUSTNESS

In this section, we introduce the proposed probing task designed to systematically examine the spatial robustness of LVLM's semantic understanding ability. Specifically, we assess whether model predictions remain consistent when key information appears in different regions of an image. We first introduce the design of our probing task, and then present experimental results to reveal the vulnerabilities of LVLMs on spatial-semantic understanding.

### 3.1 TASK DESIGN

To systematically evaluate the spatial robustness of LVLMs, we construct a probe dataset based on image–text matching. Specifically, we randomly sample 10,000 image–caption pairs $(I_m, C_m)$ from the LAION dataset (Schuhmann et al., 2022). As shown in Figure 6, for each key image $I_m$, we first randomly retrieve 8 distractor images from LAION. We then arrange $I_m$ and the 8 distractors in a $3 \times 3$ grid to form a composite image $\boldsymbol{I}_{m,0}$, which is presented to the LVLM together with the caption $C_m$. The model is asked a *yes-or-no* question $Q_m$ to determine whether any sub-image within the composite matches the given caption $C_m$. To probe sensitivity to spatial variation, we further construct augmented composites $\{\boldsymbol{I}_{m,1}, \boldsymbol{I}_{m,2}, \dots, \boldsymbol{I}_{m,8}\}$, where the original image $I_m$ is placed at different grid locations. In each case, the same question $Q_m$ is posed to the model. The final dataset contains 90,000 samples $\{\boldsymbol{I}_{m,n}, C_m, Q_m | m \in \{0, 1, \dots, 9, 999\}, n \in \{0, 1, \dots, 8\}\}$ in total. By comparing outputs across these variants, we can directly assess whether LVLMs yield consistent predictions in response to positional changes of key information, while all other visual and textual factors remain unchanged. More details are available in Appendix B.

### 3.2 RESULTS AND FINDINGS.

We evaluate five representative LVLMs on our probing dataset, including Qwen2.5-VL-7B(Bai et al., 2025), Gemma3-12B (Team et al., 2025), LLaVA-v1.6-Mistral-7B (LLaVA-v1.6) (Liu et al.,

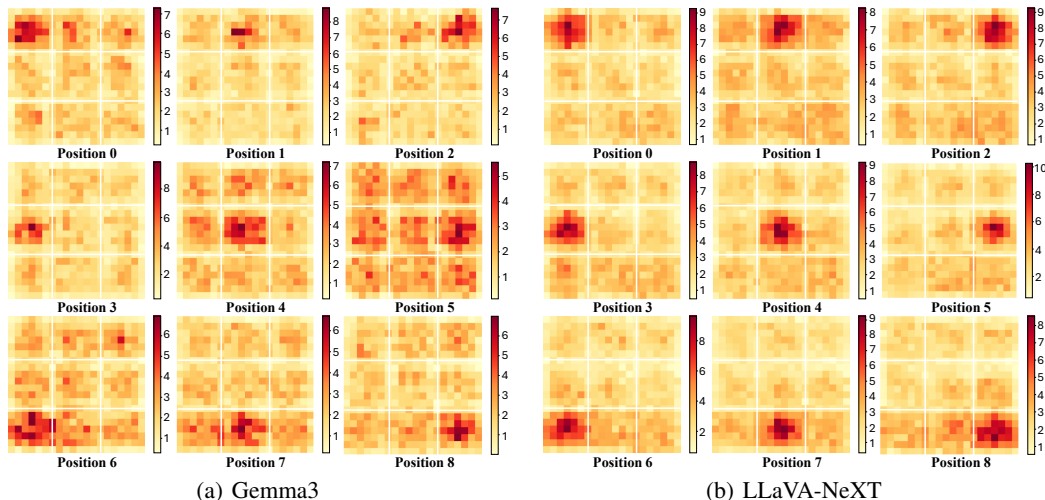

|                | (a) Gemma3 | (b) LLaVA-NeXT |
|---|---|---|

Figure 2: Results on perception ability across each *Position* $n$ of key image. Darker regions indicate higher importance scores, where logits change more significantly before and after masking. More results of Qwen2.5-VL and LLaVA-v1.6 are available in Figure 7.

2024), and Llama3-LLaVA-NeXT-8B (LLaVA-NeXT). All models are evaluated in a **zero-shot** setting, without any task-specific fine-tuning.

**Main Results.** As shown in Figure 1, all LVLMs exhibit sensitivity to the spatial variation of the key image. LLaVA-v1.6, and particularly LLaVA-NeXT exhibit strong sensitivity to spatial variations, with large fluctuations in accuracy across positions. Among all models, Qwen2.5-VL achieves the most consistent performance, likely benefiting from its improved MRoPE (Bai et al., 2025), although slight spatial bias can still be observed. Moreover, we observe that the performance is not well correlated with token distance, suggesting that spatial bias might not stem from the long-term decay property of RoPE (Su et al., 2024).

**Impact of Model Scale.** We further examine the correlation between model scale and spatial bias by evaluating the Qwen2.5-VL model series on our probe dataset. As shown in Table 1, Qwen2.5-VL-7B exhibits greater accuracy fluctuations across different grid positions ($\Delta = 1.74$), reflecting higher spatial sensitivity and stronger bias. With increasing model size, Qwen2.5-VL-32B generally achieves lower variance, which is consistent with previous observations (Zhu et al., 2024) that larger models tend to make more consistent decisions. Interestingly, the average accuracy of Qwen2.5-VL-32B declines slightly compared to Qwen2.5-VL-7B, the reason can be that larger models might be more susceptible to overfitting or inefficiencies in processing specific data types.

## 4 ANALYZING THE ORIGIN OF SPATIAL BIAS

To pinpoint the origin of the spatial bias in LVLMs, this section systematically examines the role of the vision encoder. Our analysis focuses on two critical aspects of the encoder: its low-level visual perception and high-level visual understanding capabilities, aiming to ascertain if the bias stems from this component.

### 4.1 IS VISION ENCODER PERCEPTION SPATIALLY ROBUST?

**Task Design.** Inspired by the work on model interpretability (Si et al., 2024; Zhang et al., 2025a), we design a set of experiments using eraser search (Li et al., 2016; De Cao et al., 2020) based on our probe dataset, where each region of the input image are occluded sequentially and we observe the resulting changes in the model's response. This allows us to examine how the model's behavior is affected by different spatial locations of key image $I_m$ in the composite input $\boldsymbol{I}_{m,n}$. In specific, for each position $n$, we first randomly select 20 key images, with a total of 180 samples. As shown in

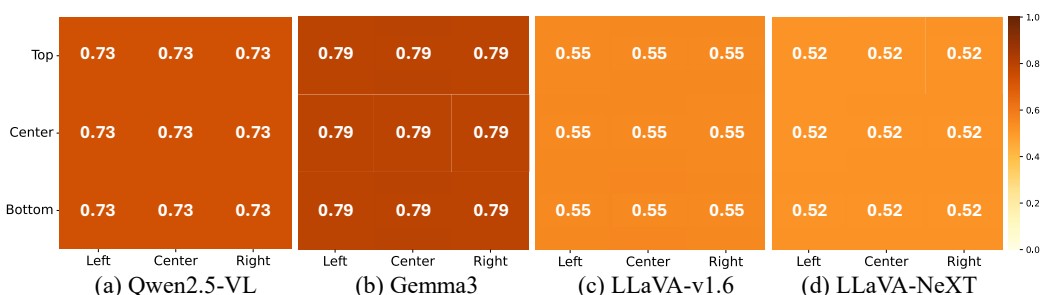

Figure 3: Results on understanding abilities across each *Position* $n$. Each cell reports the similarity between visual features and text embeddings input to the LLMs when the key image $I_m$ is placed at a specific grid location $n$ in the $3 \times 3$ composite image $\boldsymbol{I}_{m,n}$.

Figure 8(a), each composite image is divided into 400 non-overlapping regions, and we mask each region one at a time by perturbing its pixels to the background color (white in our experiment). Then, we calculate the difference between the logits of generated token based on the original composite image and the masked version for each LVLM and take it as the importance score of the region with respect to the response. Finally, we aggregate these importance scores into a heatmap to visualize how information from different positions within the input image influences the model's decision.

**Results and Findings.** Figure 2 presents the results of our masking experiments on Gemma3 and LLaVA-NeXT. In both models, the vision encoder is consistently able to identify the critical regions corresponding to the key image, irrespective of its location in the composite input. In addition, the overall patterns remain semantically aligned with the key image and do not shift when its position changes, demonstrating that the **vision encoder's perception is robust to spatial variation**. These results rule it out as the source of the spatial bias. More results are presented in Figure 7.

## 4.2 IS VISION ENCODER UNDERSTANDING SPATIALLY ROBUST?

**Task Design.** Inspired by Radford et al. (2021), we determine whether the vision encoder can maintain consistent semantic understanding when key image locations change by analyzing the similarity between the visual features input to the LLMs and the embeddings of the corresponding caption. Specifically, we first randomly select 1,000 key image-caption pairs $(I'_m, C'_m)$ from LAION dataset. As illustrated in Figure 8(b), each image $I'_m$ is then pasted onto a white background at every position $n$, serving as synthetic image inputs $\boldsymbol{I}'_{m,n}$ for the LVLMs. We measure the semantic understanding capability of the vision encoder by evaluating the cosine similarity between the embedding representations of caption $C'_m$ and visual features of $\boldsymbol{I}'_{m,n}$ input to LLM.

$$\text{Similarity} = \cos(g(f_v(\boldsymbol{I}'_{m,n})), E(C'_m)), \tag{1}$$

where $g(\cdot)$ and $f_v(\cdot)$ denote the projection module and vision encoder of the LVLM, respectively. $E(C'_m)$ is the embeddings of input caption $C'_m$.

**Results and Findings.** As shown in Figure 3, the similarity scores remain highly stable regardless of the spatial placement of the key image across all five models. This consistency indicates that the vision encoder reliably extracts semantically aligned representations of the key image, independent of its spatial location in the composite. Moreover, combining the results in Figure 1, we find a trend that LVLMs achieving higher similarity scores (e.g., Gemma3 and Qwen2.5-VL) also exhibit stronger and more stable performance on our probing tasks, whereas models with lower similarity scores (e.g., LLaVA-v1.6 and LLaVA-NeXT) perform worse and show greater sensitivity to positional changes. This correlation further validates that robust semantic alignment between visual features and textual embeddings is a key factor behind reliable multimodal understanding.

## 5 BALANCED POSITION ASSIGNMENT (BAPA)

Based on the above observation, we assume that the root of spatial bias may be traced to the imbalance inherent in the position embeddings within the LLM component of LVLMs.

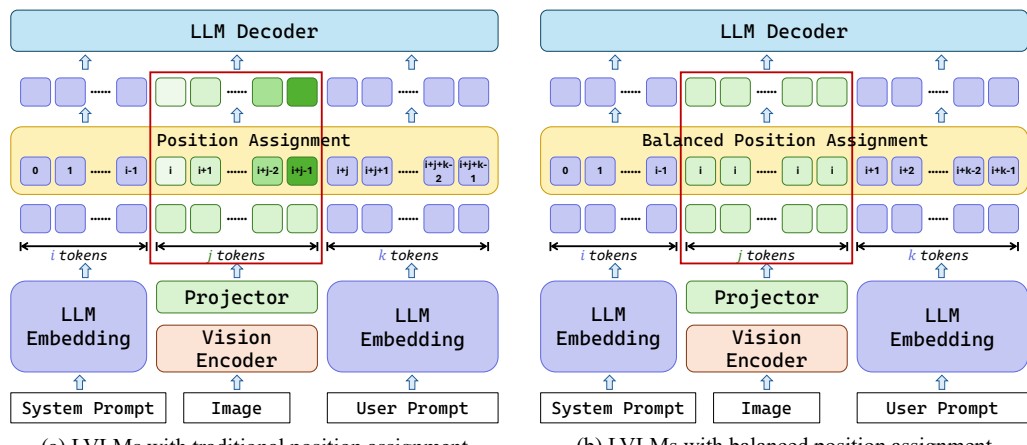

(a) LVLMs with traditional position assignment     (b) LVLMs with balanced position assignment

Figure 4: Overall view of inference in LVLMs with traditional position assignment and the proposed balanced position assignment.

## 5.1 THE BASELINE ROTARY POSITION EMBEDDING

Currently, most LVLMs employ Rotary Position Embedding (RoPE) (Su et al., 2024) as the position encoding method for the LLM component. Unlike absolute position embeddings that add a fixed vector to each token representation, RoPE encodes relative positional information directly into the attention mechanism through rotation in a complex plane. Specifically, for a token embedding $x_p$ at position $p$, RoPE applies a rotary matrix $\boldsymbol{R}_{\Theta,p}^d$ with pre-defined parameters $\Theta = \{\theta_i = 10000^{-2(i-1)/d}, i \in [1, 2, \ldots, d/2]\}$. Thus, during the self-attention computation (Vaswani et al., 2017), the dot product between query and key at position $p$ and $q$ becomes

$$q_p^{\mathsf{T}} k_q = (\boldsymbol{R}_{\Theta,p}^d W_q x_p)^{\mathsf{T}} (\boldsymbol{R}_{\Theta,q}^d W_k x_q) = x_p^{\mathsf{T}} W_q \boldsymbol{R}_{\Theta,q-p}^d W_k x_q \tag{2}$$

where $\boldsymbol{R}_{\Theta,q-p}^d = (\boldsymbol{R}_{\Theta,p}^d)^{\mathsf{T}} \boldsymbol{R}_{\Theta,q}^d$ and $q - p$ represents the relative position between $q_p$ and $k_q$. According to Equation 2, RoPE naturally introduces imbalance, where the result of the inner product, i.e., interaction strength, is influenced by the relative distance between tokens. While this imbalance is well-suited for capturing autoregressive dependencies in text, we posit that it is ill-suited for facilitating effective cross-modal interactions below.

## 5.2 THE PROPOSED METHOD

In traditional LVLMs, visual features extracted from the vision encoder are typically arranged into a 1-D sequential order following a raster-scan strategy as shown in Figure 4(a). Due to the imbalance property of RoPE (Peng et al., 2024), image tokens at different positions in the sequence have unequal influence on the same text token during cross-modal interaction. However, intuitively, since the vision encoder has already modeled local pixel relationships and global spatial structures, image tokens should be treated as semantically equivalent from the LLM's perspective.

To overcome this imbalance, we propose **Balanced Position Assignment (BaPA)**, a simple yet effective modification to the position assignment of image tokens. In particular, as shown in Figure 4(b), for a a multimodal sequence $Z = \{s_1, \ldots, s_i, \tilde{v}_1, \ldots, \tilde{v}_j, x_1, \ldots, x_k\}$, where $\{\tilde{v}_1, \ldots, \tilde{v}_j\}$ denotes the image tokens derived from vision encoder and projector, $\{s_1, \ldots, s_i\}$ and $\{x_1, \ldots, x_k\}$ denotes the embeddings of system prompt and user prompt, respectively, instead of endowing each image token $\tilde{v}_k$ with distinct position assignment $p_{\tilde{v}_k}$, BaPA enforces

$$p_{\tilde{v}_1} = p_{\tilde{v}_2} = \cdots = p_{\tilde{v}_j} = p_{img} = i \tag{3}$$

where $i$ and $j$ are the number of the system prompt tokens and image tokens, respectively. Here, in order to ensure the continuity of the overall position assignment, we set $p_{img}$ to the value of the original $p_{\tilde{v}_1}$, i.e., $i$. Therefore, the final position assignment for the entire input of the LLM backbone is as follows:

$$\text{Position\_ids} = \{\underbrace{0, 1, \ldots, i-1}_{\text{system prompt}}, \underbrace{i, i, \ldots, i}_{\text{image tokens}}, \underbrace{i+1, i+2, \ldots, i+k-1}_{\text{user prompt}}\} \tag{4}$$

Table 1: Results on our probe dataset where 0-8 denotes the different grid positions of key image in the composite images, $Avg$ and $\Delta$ denote the average accuracy and variance across all positions, respectively. The results with an increase after BaPA are **bolded**. Noted that the results of BaPA are obtained without training the baseline LVLMs.

| Position | 0 | 1 | 2 | 3 | 4 | 5 | 6 | 7 | 8 | Avg ↑ | Δ ↓ |
|---|---|---|---|---|---|---|---|---|---|---|---|
| Gemma3 | 83.77 | 83.05 | 78.11 | 85.11 | 84.43 | 78.98 | 84.41 | 82.87 | 77.60 | 0.82 | 8.75 |
| Gemma3-BaPA | 93.03 | 93.10 | 91.05 | 92.66 | 92.97 | 90.01 | 92.49 | 92.94 | 90.48 | **92.08** | **1.49** |
| LLaVA-NeXT | 68.91 | 77.62 | 70.89 | 58.54 | 69.82 | 62.52 | 55.20 | 63.89 | 57.15 | 64.95 | 54.90 |
| LLaVA-NeXT-BaPA | 96.44 | 96.35 | 96.52 | 96.31 | 96.12 | 96.47 | 96.42 | 96.46 | 96.63 | **96.41** | **0.02** |
| LLaVA-v1.6-BaPA | 75.71 | 77.75 | 71.67 | 70.70 | 65.87 | 68.35 | 65.87 | 72.40 | 68.81 | 70.82 | 16.79 |
| LLaVA-v1.6-BaPA | 94.89 | 84.47 | 93.63 | 94.18 | 93.79 | 92.71 | 94.05 | 93.77 | 92.55 | **93.78** | **9.97** |
| Qwen2.5-VL-7B | 81.86 | 80.00 | 81.67 | 83.81 | 81.71 | 83.58 | 84.15 | 82.62 | 83.05 | 82.49 | 1.74 |
| Qwen2.5-VL-7B-BaPA | 80.95 | 78.71 | 79.92 | 81.95 | 80.43 | 82.23 | 83.41 | 81.54 | 82.38 | 81.28 | 2.06 |
| Qwen2.5-VL-32B | 82.25 | 81.82 | 82.58 | 82.39 | 81.47 | 81.66 | 82.52 | 81.92 | 81.00 | 81.96 | 0.28 |
| Qwen2.5-VL-32B-BaPA | 82.31 | 81.69 | 82.31 | 82.72 | 81.66 | 81.53 | 84.18 | 83.57 | 82.34 | **82.48** | 0.80 |

This modification removes the artificial positional imbalance introduced by RoPE or similar encodings, ensuring that each image token has equal importance when interacting with textual tokens.

### 5.3 EXPERIMENTS ON PROBE DATASET

**Experimental Settings.** We first evaluate the effectiveness of the proposed BaPA on our probe dataset. Since the training data of mainstream LVLMs is not publicly available, retraining these models with a modified position embedding is infeasible. Thus, we directly apply BaPA to the inference stage of existing LVLMs without any retraining, making our approach lightweight and broadly applicable. We test the same five representative LVLMs under a zero-shot setting.

**Results and Findings.** Table 1 reports the performance of each LVLM and their BaPA-enhanced variants on our probe dataset, where the key image is placed at different grid positions (0–8). For each model, we further report the average accuracy (Avg) and variance ($\Delta$) across all positions. The results highlight three major findings.

- For **average accuracy**, applying BaPA leads to a substantial improvement across almost all models, with particularly remarkable gains observed for previous underperforming models like LLaVA-NeXT and LLaVA-v1.6. This suggests that the baseline's unbalanced position embeddings hinder the full utilization of visual information, and rectifying this imbalance enables LVLMs to exploit visual features more effectively for cross-modal interaction.

- For **spatial robustness**, BaPA effectively reduces the accuracy variance across different positions. Notably, LLaVA-NeXT, which originally suffered from severe instability ($\Delta = 54.90$), achieves near-uniform performance ($\Delta = 0.02$) after applying BaPA. This confirms that the root of spatial bias lies in the unbalanced position assignment within LLMs, and that equalizing positional importance directly mitigates this inconsistency.

- For **model scale**, as the model parameters increase from 7B to 32B, BaPA delivers a more pronounced overall accuracy improvement for Qwen2.5-VL, rising from 81.28 to 82.48. This suggests that the proposed method is effective in enhancing the performance of larger models. Moreover, owing to its improved MRoPE (Bai et al., 2025), the spatial bias inherent in Qwen2.5-VL is less significant, hence our BaPA provides no improvement on variance.

### 5.4 EXPERIMENTS ON DOWNSTREAM TASKS

**Experimental Settings.** To further validate the effectiveness of BaPA beyond the probing task, we evaluate its performance on general downstream multimodal tasks with three representative LVLMs, including Gemma3-8B (Team et al., 2025), LLaVA-v1.6-Mistral-7B (Liu et al., 2024), Qwen2.5-VL-7B (Bai et al., 2025). Considering that downstream tasks involve more complex image–text interactions than our probe dataset, we perform lightweight LoRA (Hu et al., 2022) fine-tuning with 10K instruct-tuning data from LLaVA to adapt these models to the new position encoding scheme. We additionally train LLaVA-v1.5 (Liu et al., 2024) with BaPA from scratch to compare with CCA, which is a positional alignment strategy proposed by Xing et al. (2024) to address the object hallucination of LVLMs. Following (Liu et al., 2024), we only retained samples that contain

Table 2: Results on downstream tasks. The results with an increase after BaPA are **bolded**.

| Models | MMMU-Pro$_{direct}$ | | ScienceQA$_{image}$ | CRPE$_{relation}$ | HallusionBench |
|---|---|---|---|---|---|
| | 4-option | 10-option | | | |
| **Gemma3** | 0.4357 | 0.2536 | 0.8156 | 0.7002 | 0.6393 |
| **Gemma3-BaPA** | 0.4162 | **0.2944** | **0.8180** | 0.6819 | 0.6015 |
| **Qwen2.5-VL** | 0.4763 | 0.3427 | 0.7898 | 0.7658 | 0.7066 |
| **Qwen2.5-VL-BaPA** | **0.4915** | **0.3735** | **0.8909** | **0.7690** | 0.6909 |
| **LLaVA-v1.6** | 0.3390 | 0.2015 | 0.7288 | 0.6884 | 0.5363 |
| **LLaVA-v1.6-BaPA** | 0.3365 | 0.1927 | 0.7288 | **0.6905** | **0.5489** |
| **LLaVA-v1.5** | 0.3522 | 0.1908 | 0.6951 | 0.6671 | 0.4825 |
| **LLaVA-v1.5-CCA** | 0.3340 | 0.1864 | 0.6966 | 0.6581 | 0.4921 |
| **LLaVA-v1.5-BaPA** | 0.3440 | **0.1952** | **0.7010** | **0.6760** | **0.5205** |

*exactly one image* in the input for each dataset during evaluation. More details are available in Appendix C.3.

**MMMU-Pro.** We first evaluate the proposed method on MMMU-Pro (Yue et al., 2025), which is an enhanced multimodal benchmark designed to rigorously assess the true understanding capabilities of LVLMs. As shown in Table 2, BaPA yields mixed effects on such complex reasoning tasks, while CCA typically reduces the performance of LVLMs. BaPA provides a slight decrease for Gemma3 in the 4-option setting, yet it confers a significant advantage (from 0.2536 to 0.2775) in the more challenging 10-option setting. For Qwen2.5-VL, BaPA consistently enhances performance in both settings, especially for 10-option with 3% accuracy improvement. This suggests that the benefits of BaPA are most pronounced in tasks that require richer contextual reasoning from multimodal inputs.

**ScienceQA.** We then evaluate our method on the ScienceQA dataset (Lu et al., 2022), which is collected from elementary and high school science curricula. The results in Table 2 show that BaPA can consistently improve or maintain LVLMs' performance on science-related tasks. The most notable gain is observed in Qwen2.5-VL with a substantial improvement of 10%. This suggests that BaPA effectively reduces positional imbalance, leading to better integration of visual and textual information for precise knowledge-based reasoning.

**Hallucination Benchmark.** We also evaluate BaPA on two comprehensive multimodal hallucination benchmarks, i.e., CRPE (Wang et al., 2024b) and HallusionBench (Guan et al., 2024). According to Table 2, we find that BaPA generally reduces hallucination tendencies and improves the prediction reliability of LVLMs. For instance, BaPA improves the performance of LLaVA-v1.6 on HallusionBench from 0.5363 to 0.5489, while Qwen2.5-VL achieves a small gain on CRPE. Although some models, such as Gemma3, exhibit minor drops, the overall trend suggests that BaPA helps alleviate the negative effects of spatial bias, leading to more stable and trustworthy predictions.

**MME.** The MME benchmark (Fu et al., 2023) is a comprehensive evaluation benchmark for LVLMs. Following Xing et al. (2024), we evaluate our method on four perception subtasks that assess object-level and attribute-level hallucinations. Here we also compare the results of UAC (Zhu et al., 2025c)[1], a training-free attention calibration method for object hallucination. Due to the differing training strategies of the two baselines, we employ a compromise approach for comparison, i.e., fine-tuning LLaVA-v1.5 using the BaPA with LoRA on 10K samples as other models. The results in Table 3 show that BaPA consistently preserves or improves

Table 3: Results on MME benchmark. The results with an increase after BaPA are **bolded**.

| Model | Object-level | | Attribute-level | | Total |
|---|---|---|---|---|---|
| | existence | count | position | color | |
| **Gemma3** | 190.00 | 165.00 | 98.33 | 158.33 | 611.66 |
| **Gemma3-BaPA** | **200.00** | 151.67 | **111.67** | 153.33 | **616.67** |
| **Qwen2.5-VL** | 200.00 | 155.00 | 160.00 | 195.00 | 710.00 |
| **Qwen2.5-VL-BaPA** | 200.00 | **163.33** | **175.00** | 195.00 | **733.33** |
| **LLaVA-v1.6** | 200.00 | 155.00 | 133.33 | 185.00 | 673.33 |
| **LLaVA-v1.6-BaPA** | 200.00 | 153.33 | **145.00** | 175.00 | **673.33** |
| **LLaVA-v1.5** | 175.67 | 124.67 | 114.00 | 151.00 | 565.33 |
| **LLaVA-v1.5-UAC** | 190.00 | 155.00 | 128.33 | 165.00 | 638.33 |
| **LLaVA-v1.5-CCA** | 190.00 | 148.33 | 128.33 | 175.00 | 641.66 |
| **LLaVA-v1.5-BaPA** | 190.00 | **153.33** | **135.00** | 170.00 | **648.33** |

---

[1]Here we compare with UAC rather than DAC proposed by Zhu et al. (2025c), as DAC is specifically fine-tuned on the 5.4k data that is constructed with ground truth object labels, making a direct comparison unfair.

model performance on the MME benchmark, especially at the attribute level, where position imbalance has the strongest negative effect. Compared with CCA and UAC, our BaPA achieves better results on most tasks and total scores despite being a much simpler positional adjustment. Overall, these findings indicate that equalizing image-token position embeddings is a compact, effective way to strengthen LVLMs' grounding and reduce hallucination without heavy re-training or complex architectural changes.

## 5.5 ANALYSIS OF INFORMATION FLOW

To analyze the effect of BaPA on cross-modal interactions, we visualize the information flow in LLaVA-v1.6 and the BaPA-finetuned version across three *well-performing* downstream datasets, including ScienceQA, CRPE, and HallusionBench. For each dataset, we compute the average attention score from text tokens to image tokens across all layers and attention heads, thereby highlighting the contribution of each image token during multimodal reasoning.

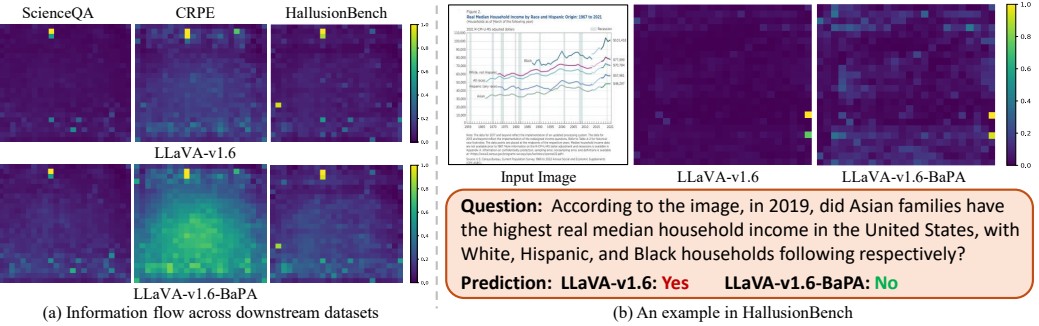

(a) Information flow across downstream datasets        (b) An example in HallusionBench

Figure 5: The visualization of information flow from image token $\tilde{v}_i$ to text token $x_j$.

**Overall Trend.** The visualization of information flow across downstream datasets shown in Figure 5(a) reveals clear differences between LLaVA-v1.6 and the BaPA-finetuned version. The baseline model tends to concentrate attention on only a few image tokens, often overlooking other semantically relevant regions. This behavior indicates that its cross-modal interaction relies heavily on localized cues, which can increase the risk of biased or insufficient interactions. In contrast, BaPA encourages a more even spread of attention across image tokens, ensuring that a broader set of important visual features contributes to the cross-modal interaction. These results demonstrate that BaPA effectively stabilizes attention patterns, allowing LVLMs to capture richer and more comprehensive visual information during cross-modal interaction.

**Case study.** To better illustrate the impact of BaPA, we present a case study from HallusionBench in Figure 5(b). The baseline LLaVA-v1.6 confines its attention to a few isolated image regions, while most visual tokens receive little to no attention. This uneven distribution reflects the position imbalance introduced by standard position assignment, leading the model to overemphasize spurious cues while neglecting relevant evidence. By contrast, LLaVA-v1.6-BaPA exhibits a more balanced allocation of attention across image tokens, particularly in regions strongly associated with correct decision—making (i.e., tokens corresponding to the curve and words). This balanced allocation facilitates the integration of visual information from broader areas, mitigates spatial bias, and thereby enhances the robustness of cross-modal understanding. More cases are available in Appendix D.

## 6 CONCLUSION

This work presents a systematic investigation of spatial bias in LVLMs, showing that inconsistent predictions under spatial shifts originate from unbalanced position encodings in the LLM. To address this issue, we propose Balanced Position Assignment (BaPA), which assigns identical position embeddings to all image tokens. Experiments demonstrate that BaPA improves spatial robustness without retraining and further enhances performance across multimodal benchmarks with lightweight fine-tuning. Further analysis demonstrates that BaPA effectively balances attention distributions, enabling more comprehensive visual understanding.

ETHICS STATEMENT

All datasets used in this work are publicly available and widely adopted in the research community. We have manually inspected the data to ensure compliance with the ICLR Code of Ethics. No offensive, harmful, or privacy-sensitive content is involved in our study.

REPRODUCIBILITY STATEMENT

To facilitate reproducibility, we have released the code and part of the data at `https://anonymous.4open.science/r/BaPA-1124/`. Meanwhile, the implementation details of each experiments are provided in Section 3.1, Section 4.1, Section 4.2, Section 5.3, Section 5.4 and Section 5.5, as well as in Appendix C.3. The complete probing dataset and checkpoints of each LVLM will be made publicly available on HuggingFace upon acceptance.

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

## A  LIMITATIONS

This paper presents a systematic investigation of spatial bias in LVLMs. Although providing new insights and solutions, our work suffers from two limitations. First, our probing experiments focus primarily on a $3 \times 3$ spatial grid, which may not fully capture more fine-grained or irregular spatial variations. Second, due to computational constraints, we limit our evaluation to medium-scale LVLMs and do not include very large models such as Qwen2.5-VL-72B. Investigating whether the identified spatial bias persists or evolves in larger-scale models remains an important direction for future work.

## B  DETAILS OF PROBE TASK

For each key image $I_m$ and caption $C_m$ randomly selected from LAION, we first generate 9 corresponding composite images $\{I_{m,n}\}_{n=0}^{8}$ according to the workflow depicted in Figure 6. Then we fed each composite image $I_{m,n}$ alongside the question $Q_m$ format as following into the LVLMs to test their spatial robustness.

> **Question $Q_m$**
>
> Determine if there is a sub-image in the given image that matches the text following.
> Text: $\{C_m\}$
> The answer should only contain 'Yes' or 'No', without reasoning process.

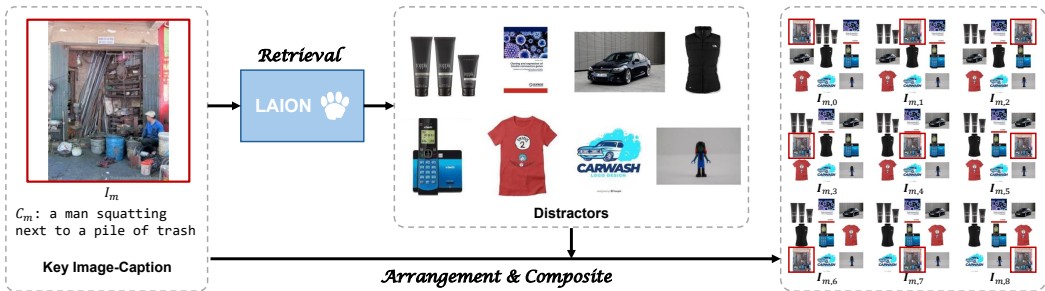

Figure 6: Workflow illustration on how we synthesize composite images in probe dataset, where the key image features red borders not present in our experiments.

## C  EXPERIMENTS ON DOWNSTREAM TASKS

### C.1  DATASETS

**MMMU-Pro**  MMMU-Pro (Yue et al., 2025) is an enhanced multimodal benchmark designed to challenge and evaluate multimodal models with tasks demanding college-level subject knowledge and complex reasoning. It contains 1.73K meticulously collected multimodal questions from college exams, quizzes, and textbooks, covering six core disciplines: Art & Design, Business, Science, Health & Medicine, Humanities & Social Science, and Tech & Engineering. To facilitate statistical analysis, we evaluate each LVLM under 4-option and 10-option settings, respectively, and ask them to answer questions directly.

**ScienceQA**  ScienceQA (Lu et al., 2022) is a large-scale multi-choice dataset collected from elementary and high school science curricula, and contains 21,208 multimodal science questions with explanations and features rich domain diversity. We assess each LVLM on the test set of ScienceQA with 4,241 samples.

**CRPE**  CRPE is a benchmark designed to quantitatively evaluate the object recognition and relation comprehension ability of LVLMs. The evaluation is formulated as single-choice questions.

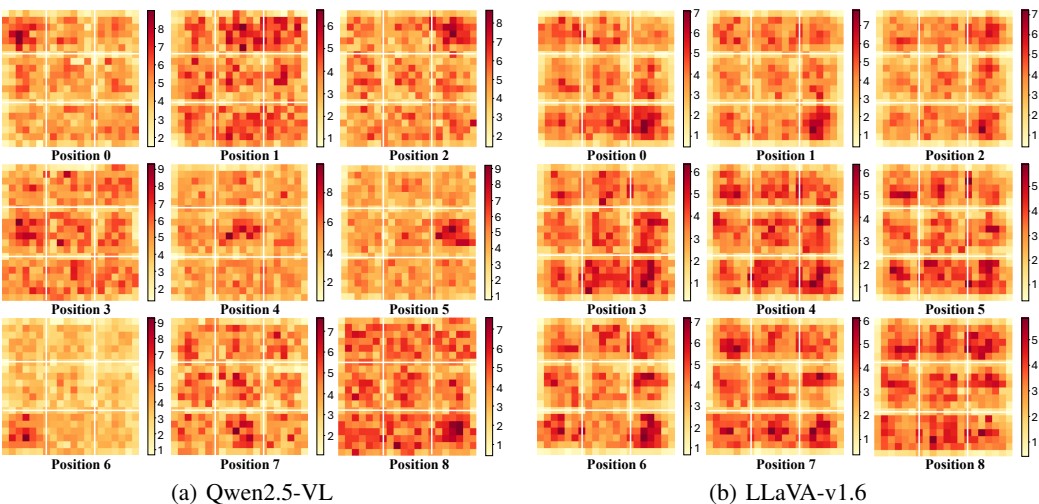

(a) Qwen2.5-VL      (b) LLaVA-v1.6

Figure 7: More results on perception ability across each *Position* $n$ of key image. Darker regions indicate higher importance scores, where logits change more significantly before and after masking.

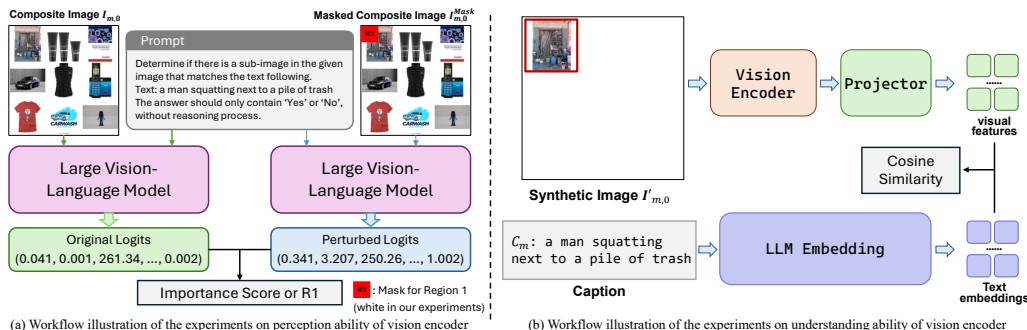

(a) Workflow illustration of the experiments on perception ability of vision encoder      (b) Workflow illustration of the experiments on understanding ability of vision encoder

Figure 8: Workflow illustration on our analysis experiments.

Following previous work (Bai et al., 2025), we evaluate LVLMs on relation comprehension ability instead of recognition with CRPE.

**HallusionBench** HallusionBench is a comprehensive benchmark designed for the evaluation of image-context reasoning, which comprises 346 images paired with 1129 questions, all meticulously crafted by human experts.

## C.2 BASELINES

**CCA** Concentric Causal Attention (CCA) (Xing et al., 2024) is a simple yet effective positional alignment strategy designed for mitigating object hallucination in LVLMs. The core novelty of CCA lies in naturally reducing relative distance between visual and instruction tokens, thereby mitigating the long-term decay effects of RoPE in LVLMs. CCA trains LLaVA from scratch through two stages to adapt the new position embedding, including 1) a pre-training over CC-558K dataset with global batch size of 256 and 2) a instruction tuning with a 665k multi-turn conversation dataset with global batch size of 128. During the experiment, we directly use the checkpoints provided by Xing et al. (2024) for evaluation[2].

**UAC** Uniform Attention Calibration (UAC) (Zhu et al., 2025c) is a training-free method that removes spatial perception bias estimated from a meaningless input by calibrating biased attention,

---

[2]https://huggingface.co/xing0047/cca-llava-1.5-7b

Table 4: Hyperparameters for LoRA finetuning.

| Model | Lora_rank | Lora_alpha | Global Batch Size | Learning rate | Epoch |
|-------|-----------|------------|-------------------|---------------|-------|
| **Gemma3** | 8 | 16 | 32 | 1.0e-04 | 2 |
| **LLaVA-v1.6** | 8 | 16 | 32 | 1.0e-04 | 1 |
| **Qwen2.5-VL** | 8 | 16 | 32 | 1.0e-04 | 1 |
| **LLaVA-v1.5** | 128 | 256 | 128 | 2.0e-04 | 1 |

Table 5: Hyperparameters for training LLaVA-v1.5 from scratch.

| Stage | Global Batch Size | Learning rate | Epochs | Max length | Weight decay |
|-------|-------------------|---------------|--------|------------|--------------|
| Pre-training | 256 | 1.0e-03 | 1 | 2048 | 0 |
| Instruct-tuning | 128 | 2.0e-05 | 1 | 2048 | 0 |

offering a simple yet effective solution with competitive performance. Due to the unavailability of the relevant code and models, we did not compare our method with UAC on general downstream tasks. Instead, we only report its results on the MME benchmark provided by Zhu et al. (2025c).

### C.3 IMPLEMENTATION DETAILS

For each LVLM, we randomly select 10,000 samples from the LLaVA-v1.5 instruction-tuning data[3] and fine-tune the models with LoRA (Hu et al., 2022) using the LLaMA-Factory library[4]. The hyperparameters used for training are shown in Table 4. To better adapt LLaVAv1.5 to BaPA, we retrain it from scratch using the same two stages as (Xing et al., 2024), and the hyperparameters are presented in Table 5 All the experiments are finished on 4 A100 GPUs with 80GB memory.

## D CASE STUDY

We further randomly select one example each from HallusionBench, ScienceQA and CRPE for demonstration. The results are shown in Figures 9, 10 and 11.

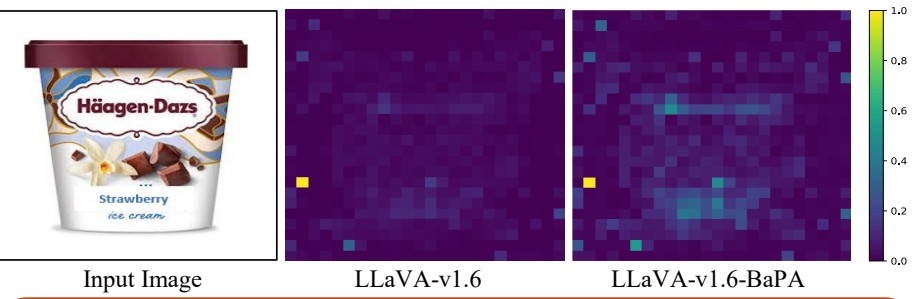

Input Image       LLaVA-v1.6       LLaVA-v1.6-BaPA

**Question:** According to the text given in the image, is this ice-cream a vanilla-chocolate flavor ice cream?
**Prediction: LLaVA-v1.6: Yes**      **LLaVA-v1.6-BaPA: No**

Figure 9: An example in HallusionBench, where the input image is resized as processed in LLaVA-v1.6.

---

[3] https://huggingface.co/datasets/liuhaotian/LLaVA-Instruct-150K/blob/main/llava_v1_5_mix665k.json
[4] https://github.com/hiyouga/LLaMA-Factory

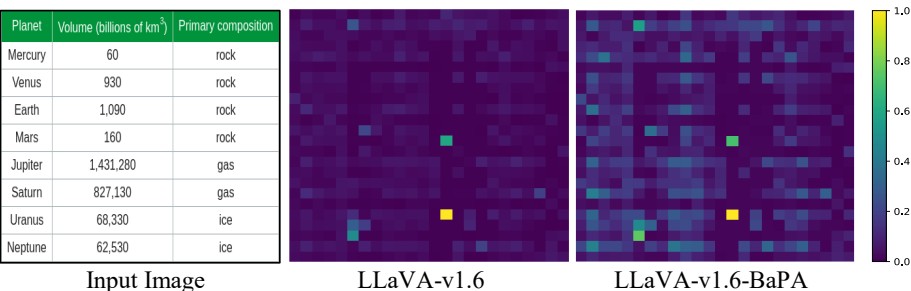

Context: Use the data to answer the question below.
Is the following statement about our solar system true or false?
Neptune's volume is more than 50 times as great as that of Earth.
A. true     B. false

**Prediction: LLaVA-v1.6: B       LLaVA-v1.6-BaPA: A**

Figure 10: An example in ScienceQA, where the input image is resized as processed in LLaVA-v1.6.

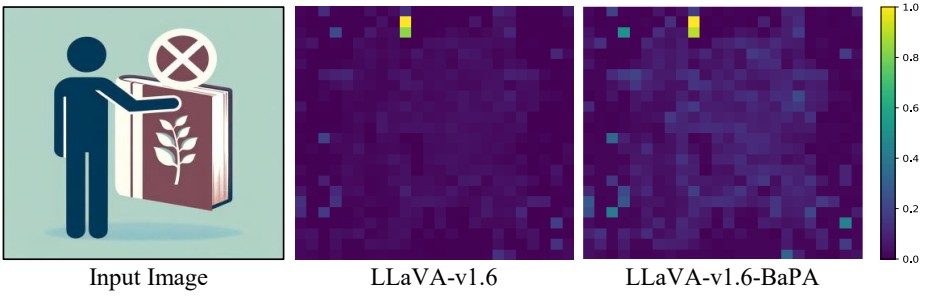

**Question:** What is the relation between the person and the book?
A. The person is beside the book. B. The person is sitting on the book.
C. The person is exiting the book. D. The person is standing on the book.

**Prediction: LLaVA-v1.6: B       LLaVA-v1.6-BaPA: A**

Figure 11: An example in CRPE, where the input image is resized as processed in LLaVA-v1.6.

## E   THE USE OF LARGE LANGUAGE MODELS

During the preparation of the manuscript, we employed large language models (LLMs) to correct grammatical errors and typos to improve the fluency and readability of the text. All research ideas, experimental designs, analyses, and conclusions are solely developed by the authors.

