# OpenReview forum: "From Bias to Balance: Exploring and Mitigating Spatial Bias in LVLMs"
_ICLR.cc/2026/Conference — ICLR 2026 Conference Withdrawn Submission_

### Official Review · Reviewer_MZF9 · 2025-10-29

**Soundness:** 2
**Presentation:** 2
**Contribution:** 1
**Rating:** 2
**Confidence:** 5

**Summary:**

The paper investigates the spatial bias of Large Vision-Language Models (LVLMs), showing that models often produce inconsistent predictions when identical visual content appears in different locations. The authors attribute this bias to imbalanced positional embeddings in the LLM component and propose Balanced Position Assignment (BaPA), which assigns identical positional IDs to all image tokens. Applied during inference without retraining, BaPA improves spatial robustness and slightly enhances downstream performance across several LVLMs and multimodal benchmarks.

**Strengths:**

The paper follows a commendable research style of identifying a specific problem, systematically probing its underlying causes, and designing a targeted solution. The overall organization—from constructing a probing dataset, analyzing the vision and language components separately, to proposing a mitigation strategy—is clear and methodical. The inclusion of multiple LVLM backbones (e.g., Qwen2.5-VL, LLaVA-v1.6, Gemma3) makes the study comprehensive.

**Weaknesses:**

1. Some of the conclusions are not rigorously supported by the provided probing experiments. For instance, Qwen2.5-VL-7B already exhibits relatively stable spatial robustness on the proposed probe task, and applying BaPA to it even slightly worsens its performance. Moreover, the claim that “the vision encoder understanding is spatially robust” is not directly validated with the proposed task, despite being plausible. The conclusion that “the root of spatial bias lies in the imbalance of positional embeddings within the LLM” also appears oversimplified, as other contributing factors, such as the attention-sink effect, are not examined.

2. The proposed positional encoding modification, Balanced Position Assignment (BaPA), is conceptually straightforward and not novel—it has already been used or discussed in prior works such as CogVLM, which similarly assigns a shared positional ID for all image tokens. In addition, the paper lacks comparisons with other recent RoPE variants designed for LVLMs, including V2PE, MRoPE, and CircleRoPE, making it difficult to assess the relative merit of BaPA.

3. The proposed positional encoding scheme may also be suboptimal given the design of current top-performing LVLMs. Modern architectures such as Qwen2.5/3-VL, KiMi-VL, and LLaVA-OV-1.5 adopt resolution-flexible schemes where 2D RoPE is applied in the vision encoder with minimal positional information propagated to the language component. Since these models benefit from letting the LLM also perceive coarse spatial structures, simply removing positional differentiation among image tokens—as BaPA does—might not align with this evolving design trend.

4. The experimental evaluation should be broader. The paper mainly includes general multimodal benchmarks like ScienceQA, MME, and MMMU-Pro, but misses fine-grained grounding and structural visual understanding benchmarks such as ChartVQA and DocVQA, which are more sensitive to positional or structural reasoning. Including such benchmarks would provide a fuller validation of spatial robustness improvements.

**Questions:**

Why is the reported score of Qwen2.5-VL on ScienceQA-Img only 0.78? My evaluation using lmms-eval is 0.87.

---

### Official Review · Reviewer_2vXx · 2025-10-30

**Soundness:** 3
**Presentation:** 3
**Contribution:** 2
**Rating:** 2
**Confidence:** 4

**Summary:**

The paper focus on benchmarking and mitigating of spatial bias. For benchmarking, they design a **probing** dataset, and find that current LVLMs is sensitive to spatial location shifts of objects. To this end, they propose Balanced Position Assignment (BaPA), a method that applies identical position embeddings to all image tokens. Such approach enhances the spatial robustness of LVLMs without retraining and boosts their performance on multiple multimodal benchmarks.

**Strengths:**

+ Topic. The study of spatial robustness is interesting.
+ Presentation. The presentation and core idea is overall easy to follow.
+ Experiment. Experiments are sufficient to support claims.

**Weaknesses:**

+ Novelty. The novelty concerns are from two aspects, including,
    - Benchmarking. CCA [Xing et al. 2024] already proposed a spatial probing data for evaluating spatial robustness, which authors ignored to mention.
    - Approach. It should be pointed out that assigning the same positions to visual tokens is not a new approach, while this paper ignores to discuss. [A]

+ Relationship between better spatial robustness and LVLM benchmark accuracy. Authors are suggested to include more deeper insights, especially how better spatial robustness benefits existing LVLMs, from what aspects LVLMs are benefited. For example, spatial relationship, referential, in what cases such correlations are strong, and in what cases such correlations are weak?

[A] Vista-LLaMA: Reducing Hallucination in Video Language Models via Equal Distance to Visual Tokens. CVPR 2024.

**Questions:**

See weakness.

---

### Official Review · Reviewer_p9cf · 2025-10-31

**Soundness:** 3
**Presentation:** 3
**Contribution:** 3
**Rating:** 2
**Confidence:** 4

**Summary:**

This paper aims to reduce the spatial bias in VLMs by introducing same position embedding values for image tokens. The authors analyses the phenomena of position bias in VLM semantic understanding by varying position of key visual information in an image, revealing that VLM  performances change as visual information shifts to different positions. Based on these findings, the authors offer a different view contrary to previous researchers that spatial bias is not a result of long-term dependency of RoPE. Following this finding, the authors proposed a position encoding modification strategy that sets the position embedding to be the same for all visual tokens. Experimental results on multiple image QA benchmarks show improved performance after applying BaPA.

**Strengths:**

Overall, the manuscript is clearly written. The issue of VLM spatial bias is clearly presented through a series of extensive analysis experiments, showcasing the effect of changing key visual information positions on model performance using multiple baseline models. The visualisations provided in figures 1,2 and 3 also offer a clear view of the spatial bias phenomena which greatly assists reader in understanding the concept.

**Weaknesses:**

This work challenges previous findings that VLM positional bias may not arise due to the long-term decay effect of RoPE. The authors further backup this claim with experiments, showing that model results are not directly correlated with distance as key visual feature shifts position. However, the authors do not specify the token length for this probing experiment. The claim can be further validated if the same experimental results are observed on high resolution data, where the input tokens are longer and long-term decay would be potentially more significant.

Though the manuscript is clearly written and easy to follow, there appear to be typos in important result sections of the paper (see questions for more detail), which may confuse the reader and undermine credibility of this work.

**Questions:**

1. In section 4.1 the authors derive the conclusion that "vision encoder’s perception is robust to spatial variation"(line 241). However, the experimental result presented in figure 2 is obtained from "logits of generated token"(line 233) for each LVLM. How does this result support the claim if it is under the influence of both vision encoder and attention layers of VLM?
2. How does BaPA affect temporal dependency on video understanding tasks when applied to video models such as Qwen 2.5 VL?
3. In table 1 under line 330, the average result for Gamma3 is reported as 0.82. Is it a typo?
4. In table 1 line 333-334, LLaVA-v1.6-BaPA is listed twice but with different results. Is it also a typo?
5. In table 1, the result of LLaVA-NeXT is significantly lower compared to other models and the LLaVA-v1.6 variant. However, in some conventions, they are the different names for the same model. Could you please specify the exact model used to avoid confusion? Additional evidence such as evaluation logs would also be welcomed to increase the credibility of this result.
6. The authors evaluated BaPA on general image QA benchmarks such as MMMU-Pro and ScienceQA. How does BaPA perform on hallucination benchmarks such as POPE and CHAIR?

---

### Official Review · Reviewer_UrNU · 2025-10-31

**Soundness:** 2
**Presentation:** 3
**Contribution:** 2
**Rating:** 4
**Confidence:** 4

**Summary:**

This paper investigates spatial bias in LVLMs—prediction inconsistency when key visual evidence is moved across image regions.  The authors trace the root cause to RoPE-induced relative-position imbalance in the LLM rather than the vision encoder.  They propose Balanced Position Assignment (BaPA), which assigns the same position id to all image tokens while preserving sequence continuity, thereby equalizing their influence on text tokens; BaPA is applied at inference without retraining.

**Strengths:**

- Clear causal diagnosis with a principled fix. The paper pinpoints RoPE-induced relative-position imbalance in the LLM, which is unsuited for cross-modal interaction. They proposes BaPA, assigning an identical position id to all image tokens while preserving sequence continuity; this is motivated by the vision encoder already modeling spatial structure.

- Rigorous and convincing evidence. A 90k-sample probe quantifies spatial robustness; BaPA consistently boosts accuracy across multiple LVLMs and sharply reduces variance.

**Weaknesses:**

- Limited novelty. The core technique (assigning one shared position id to all image tokens) was already used in CogVLM (arXiv 2023; NeurIPS 2024), which explicitly states that all visual tokens “share a single position id” under RoPE. MammothModa (2024) further applies the same principle at the frame level (“Shared Frame Position ID”) for video inputs.

- Inconsistent effects on strong base models (Qwen2.5-VL). On the probe dataset (Table 1), BaPA yields negligible or negative changes for Qwen: 7B Avg 82.49 → 81.28 (variance 1.74 → 2.06), 32B 81.96 → 82.48 (variance 0.28 → 0.80). Yet on downstream tasks (Table 2) results are mixed: ScienceQA rises 0.7898 → 0.8909 while HallusionBench drops 0.7066 → 0.6909; MMMU-Pro shows small gains. This divergence questions how reliably BaPA translates probe-level “debiasing” into broad task improvements.

1. CogVLM: Visual Expert for Pretrained Language Models，2311.03079
2. MammothModa: Multi-Modal Large Language Model， 2406.18193

**Questions:**

Please refer to the Weaknesses.

---

### Note · Authors · 2025-12-04

I have read and agree with the venue's withdrawal policy on behalf of myself and my co-authors.